# The association between epicardial adipose tissue thickness and diabetes mellitus, hyperlipidemia, hepatosteatosis, pancreatic steatosis and pancreatitis

Ece Zengin[1], Aybuke Ucgun[1], Mehmet Emir Çevik[2], Sehnaz Evrimler[1]*,
Ihsaniye Suer Dogan[1]

1 Department of Radiology, Ankara Etlik City Hospital, Ankara, Turkey, 2 Department of Radiology,
Kırıkkale University Faculty of Medicine, Kırıkkale, Turkey

* drsehnaz@gmail.com

## Abstract

### Background

Epicardial adipose tissue (EAT) is associated with cardiometabolic disorders such as diabetes mellitus (DM), hyperlipidemia, and nonalcoholic fatty liver disease. However, its potential relationship with pancreatic steatosis and pancreatitis remains unclear, and existing studies offer inconsistent findings. Therefore, a clearer understanding of whether EAT reflects broader systemic ectopic fat burden or inflammatory processes is needed.This study evaluated the relationships between EAT thickness and DM, hyperlipidemia, hepatosteatosis, pancreatic steatosis, and pancreatitis.

### Methods

This retrospective, single-center study included 200 patients who underwent abdominal CT between 2022 and 2024. EAT thickness was measured at the mid-RCA and LAD levels, and subcutaneous fat was measured at the umbilical level. Liver and pancreatic steatosis were assessed with CT or MRI. Demographic and clinical data (age, gender, LDL cholesterol, diabetes, and history of pancreatitis) were collected. Mann-Whitney U, Spearman correlation, and logistic regression were used in analyses; p < 0.05 was considered significant.

### Results

Of the 200 patients, 31.4% had diabetes, 42% had hepatosteatosis, and 73.5% had a history of pancreatitis. EAT and subcutaneous fat were significantly higher in women at all levels (p < 0.05). Diabetes was associated with increased EAT in both the RCA (p = 0.002) and the LAD (p = 0.001). In multivariate analysis, RCA-EAT (OR=1.18, p = 0.002) and age (OR=1.03, p = 0.003) were significantly associated with diabetes.

**Data availability statement:** Data cannot be shared publicly because of patient confidentiality. Data are available from the Etlik City Hospital Ethics Committee etliksh.etikkurul@saglik.gov.tr for researchers who meet the criteria for access to confidential data.

**Funding:** The author(s) received no specific funding for this work.

**Competing interests:** No authors have competing interests.

High LDL was associated with LAD-EAT (p = 0.030). For pancreatitis, multivariate analysis identified pancreatic steatosis (OR=5.78, p < 0.001) and LAD-EAT (OR=1.52, p = 0.002) as variables significantly associated with a history of pancreatitis.

## Conclusions

EAT thickness is significantly associated with DM, LDL cholesterol, pancreatitis history, and age, supporting its role as a potential imaging biomarker of cardiometabolic risk. These findings suggest that EAT may serve as an imaging marker of broader metabolic and inflammatory burden, supporting its relevance for cardiometabolic risk assessment.

## Introduction

Epicardial adipose tissue (EAT) represents a metabolically active visceral fat layer located between the myocardium and the visceral layer of the pericardium, mainly in the atrioventricular and interventricular grooves and surrounding the coronary arteries. Increased EAT thickness has emerged as an independent indicator in the development of cardiovascular diseases. In obesity, insulin resistance, diabetes mellitus (DM), and vascular injury, EAT promotes inflammatory cell recruitment to the arterial wall, fostering a proinflammatory environment and chronic vascular remodeling [1]. Growing evidence also links EAT with obesity, hepatosteatosis, DM, and LDL cholesterol [2], highlighting its relevance beyond cardiovascular disease.

Hepatosteatosis (HS), considered a marker of total body fat, is part of the non-alcoholic fatty liver disease (NAFLD) spectrum and can be quantified with high accuracy using multi-detector computed tomography (MDCT). Yet, the association between EAT and HS remains inconclusive [3]. In diabetes, EAT correlates with obesity, fasting glucose, insulin resistance, and adiponectin levels, and is consistently elevated in both Type 1 and Type 2 DM [1]. Furthermore, greater EAT thickness has been associated with higher LDL cholesterol concentrations in hyperlipidemic patients, underscoring its dual impact on cardiovascular pathogenesis and lipid metabolism [4].

Excess fat accumulation in other visceral depots also contributes to metabolic disease. Pancreatic fat deposition impairs insulin secretion and aggravates metabolic dysfunction [5], while visceral adiposity is a known determinant of severity and prognosis in acute pancreatitis [6]. Although EAT is anatomically located around the heart, it functions as an active visceral fat depot that secretes proinflammatory cytokines (IL-6, TNF-α, MCP-1) and strongly correlates with total visceral adipose tissue. This shared inflammatory and metabolic pathway makes it biologically plausible that increased EAT thickness may reflect a broader systemic ectopic fat burden, including liver and pancreatic fat accumulation. However, the relationships between EAT and pancreatic fat or pancreatitis history have not been systematically examined, and existing studies provide inconsistent findings regarding its association with hepatosteatosis. This gap limits our understanding of whether EAT represents only a cardiac

fat marker or a more general indicator of metabolic organ involvement. Therefore, we aimed to investigate the associations between EAT thickness and hepatosteatosis, diabetes mellitus, LDL cholesterol, pancreatic fat, pancreatitis history, and subcutaneous abdominal fat. We hypothesized that greater EAT thickness would be positively associated with these metabolic parameters, reflecting its potential role as a marker of systemic ectopic fat accumulation.

## Methods

This is a single-center, institutional review board approved study. This retrospective study was approved by the Ankara Etlik City Hospital Scientific Research Evaluation and Ethics Committee. Written informed consent was obtained from all participants for the use of their anonymized medical records for research purposes. All participants were aged over 18 years, and all data were fully anonymized prior to analysis with no access to identifying personal information during or after data collection. Epicardial fat tissue thickness was measured in patients who were admitted to our hospital as outpatients or inpatients and underwent abdominal CT/MRI between 1 December 2022 and 1 December 2024, and the relationship between pancreatic fat, the presence of DM, the presence of high LDL cholesterol, hepatosteatosis, and pancreatitis attacks were evaluated retrospectively. Given that one of the predefined objectives of this study was to evaluate the association between EAT thickness and pancreatitis, patients undergoing abdominal CT/MRI due to a clinical suspicion of pancreatitis were purposefully included in the study cohort. Patients were included if imaging quality was sufficient and if clinical data regarding diabetes mellitus (DM), LDL cholesterol, hepatosteatosis, subcutaneous abdominal fat, and pancreatitis history were available; those with prior cardiac surgery altering pericardial anatomy, malignancy involving the liver or pancreas, severe motion artifacts, or missing clinical or laboratory information were excluded. We accessed the medical records between 01/02/2025 and 06/05/2025 for data extraction.

Epicardial fat tissue thickness was measured by calculating the maximum thickness perpendicular to the heart surface from the myocardium to the visceral pericardium, measured in one axial plane, corresponding to the middle portion of the right coronary artery (RCA) and the lateral aspect of the left anterior descending coronary artery (LAD) (Fig 1). HS was calculated based on liver HU values obtained by drawing a region of interest (ROI) in patients undergoing CT. Hepatosteatosis was defined as a liver attenuation of at least 10 HU lower than the spleen attenuation on unenhanced CT, a liver attenuation of less than 40 HU, or a liver attenuation of 15 HU lower than the spleen (Fig 2). On contrast-enhanced CT, a liver-spleen differential attenuation (liver HU–spleen HU) in the portal venous phase of −20 HU to −43 HU, depending on

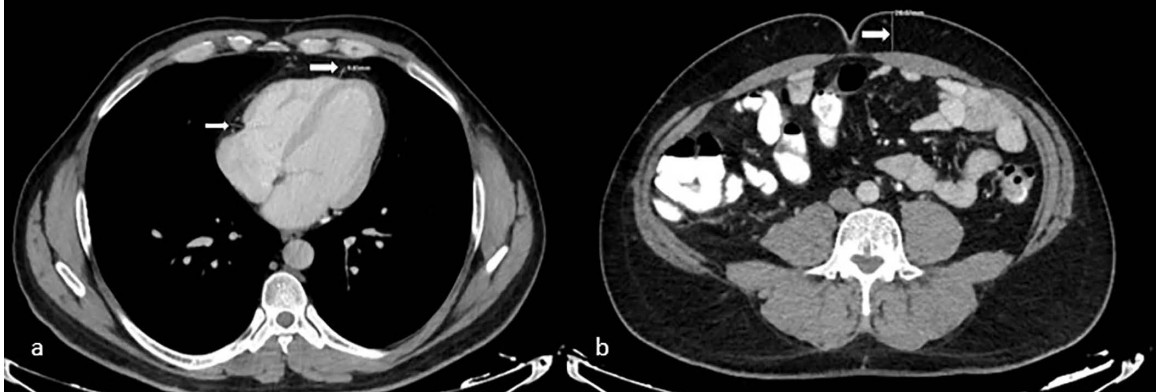

**Fig 1. a. Epicardial adipose tissue thickness was measured by calculating the maximum thickness perpendicular to the surface of the heart, from the myocardium to the visceral pericardium, at a single axial section level—specifically at the atrioventricular groove at the mid-third of the right coronary artery (RCA) and adjacent to the left anterior descending coronary artery (LAD). B.** Umbilical fat tissue thickness was measured at umbilicus level perpendicular to rectus muscle anteriorly.

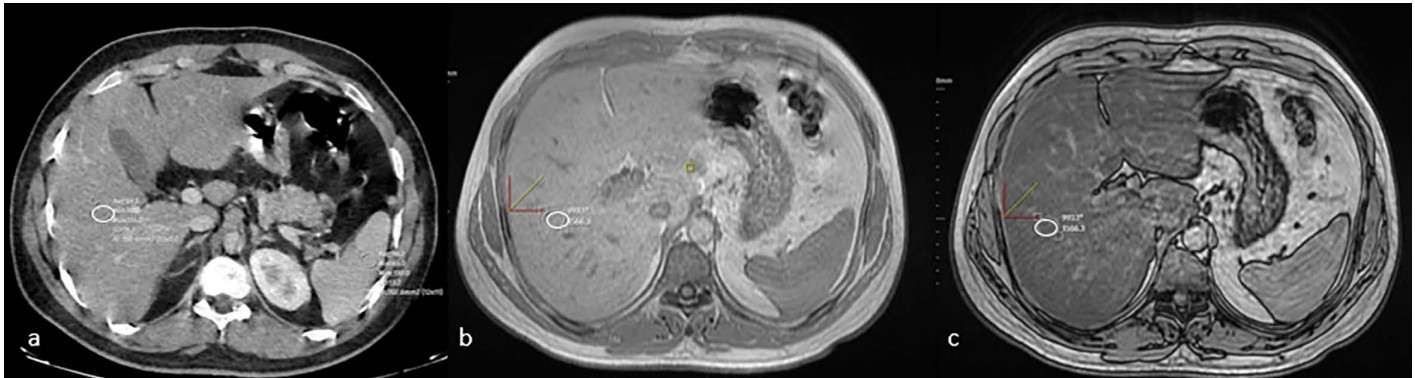

**Fig 2. a. Hepatic steatosis was assessed by measuring liver attenuation values (in Hounsfield Units, HU) on CT scans using regions of interest (ROIs) drawn within the liver parenchyma.** b and c. On dual-echo sequences, the out-of-phase image (c) demonstrates a signal reduction relative to the in-phase image (b).

the injection protocol, was considered to be the presence. In patients with available MRI, hepatosteatosis was categorized based on the presence of fat suppression on chemical shift (in-out of phase) sequences. The amount of pancreatic fat was calculated using density measurements from the head, body, and tail of the pancreas on CT scans, and subjective categorization of fat was also performed (Fig 3). Measurements were performed by two radiologists. Because ectopic fat distribution may be influenced by demographic characteristics, age and sex were recorded for all participants to allow appropriate interpretation of potential confounding effects.

## Statistical analysis

All analyses were conducted using IBM SPSS Statistics, version 26 (IBM Corp., Armonk, NY, USA). Continuous variables were summarized as mean±standard deviation (SD) and median (range), while categorical variables were expressed as frequency (n) and percentage (%). The Shapiro–Wilk test was applied to assess the normality of data distribution. For comparisons between two independent groups with non-normally distributed data, the Mann–Whitney U test was utilized. Associations between continuous variables were evaluated using Spearman's rank correlation. To explore factors related to clinical outcomes, both univariate and multivariate binary logistic regression models were performed. Variables with a

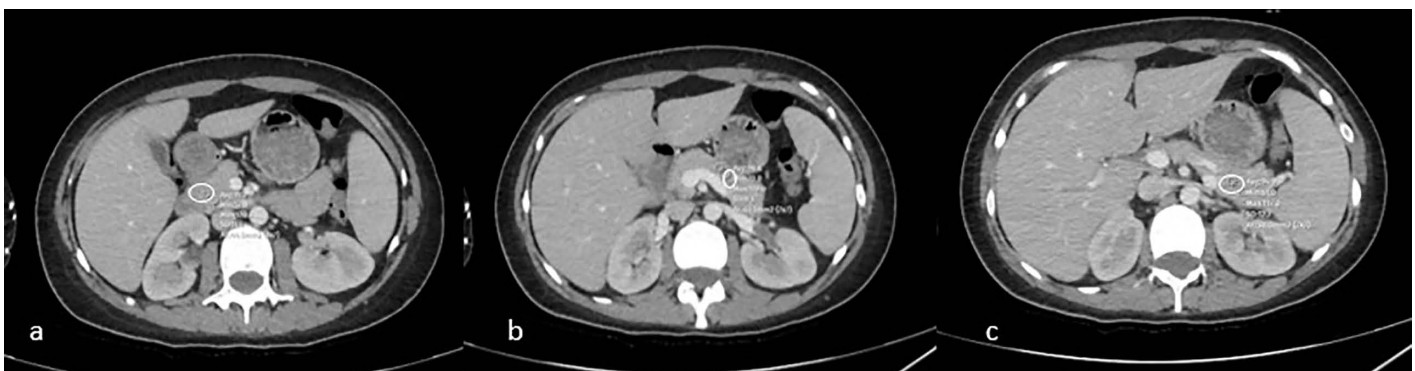

**Fig 3. Pancreatic fat content was also assessed on CT by measuring attenuation values in the head (a), body (b), and tail (c) regions of the pancreas.**

significance level of p < 0.05 in the univariate analysis were entered into the multivariate model.. All measured variables were included in the univariate regression analyses, and only those that reached statistical significance were subsequently carried forward into the multivariate model to avoid overfitting. Age and sex were retained in the model regardless of univariate significance because they represent established clinical confounders influencing adipose tissue distribution and metabolic outcomes. Logistic regression results were presented as odds ratios (OR) with 95% confidence intervals (CI). A p-value below 0.05 was considered statistically significant in all analyses.

## Results

### Study population characteristics

The study population consisted of 200 retrospectively evaluated patients. The mean age of the participants was 54.69 ± 16.61 years, and 54.5% were female (n = 109) and 45.5% were male (n = 91). LDL cholesterol levels were found to be high in 59.6% of the participants (n = 84), and normal in 40.4% (n = 57).

Epicardial adipose tissue (EAT) thickness measurements performed on CT scans revealed a median thickness of 3.01 mm (min: 1.03 – max: 10.40 mm) at the LAD level and a median of 9.44 mm (min: 4.06 – max: 27.17 mm) at the RCA level. The median subcutaneous adipose tissue thickness measured at the umbilicus level was 24.44 mm (min: 4.31 – max: 62.12 mm).

The median density values measured from the head, body and tail sections of the pancreatic tissue were found to be 52.5 HU in the head section, 54 HU in the body section and 55 HU in the tail section.

Overall, 31.4% of the participants had a history of diabetes mellitus (DM) (n = 59), 42% had a history of hepatosteatosis (n = 84), and 73.5% (n = 147) had a history of pancreatitis.

### EAT thickness and sex differences

In the comparison between genders, epicardial fat thickness was found to be significantly higher in females than in males at all measurement levels. At the RCA level, the median thickness was 8.84 mm in males and 9.92 mm in females (p = 0.001). At the LAD level, these values were 2.91 mm and 3.09 mm, respectively (p = 0.017). At the umbilicus level, the median subcutaneous fat tissue values were found to be 20.61 mm and 29.54 mm, respectively, with statistical significance (p < 0.001; see Table 1).

### EAT and diabetes mellitus

A significant difference was found between the presence of DM and EAT thickness. EAT thickness measured at the RCA level was greater in those with DM (median: 10.44 mm vs. 9.11 mm; p = 0.002). A similar significant difference was observed at the LAD level (p = 0.001). Subcutaneous adipose tissue thickness at the umbilical level did not differ significantly between patients with and without DM (p = 0.531; Table 1). In univariate and multivariate logistic regression analysis, LAD- and RCA-EAT thickness were significant univariate predictors of DM. In multivariate analysis, only age (OR = 1.03) and RCA-EAT (OR = 1.18) remained significantly associated with diabetes (Table 2).

### EAT and Hepatosteatosis.

There was no statistically significant association between hepatosteatosis and either EAT thickness or subcutaneous fat at the umbilical level (p > 0.05) (Table 1).

### EAT and LDL cholesterol

The relationship between LDL values and EAT distributions of LAD levels was statistically significant (p = 0.030). However, there was no significant difference in RCA-EAT thickness and subcutaneous fat thickness between participants with normal or elevated LDL levels (p = 0.235 and 0.290, respectively) (Table 1).

**Table 1. Comparison of EAT and subcutaneous fat thickness across key metabolic and clinical variables.**

| Variables | | EAT thickness at RCA level (median(min-max)) | p value | EAT thickness at LAD level (median(min-max)) | p value | Subcutaneous fat at umbilical level (median(min-max)) | p value |
|---|---|---|---|---|---|---|---|
| Sex | | | | | | | |
| | Male | 8.84(4.06-16.49) | **0.001** | 2.91 (1.03-6.71) | **0.017** | 20.61(4.31-57.97) | **<0.001** |
| | Female | 9.92(4.29-27.17) | | 3.09 (1.25-10.40) | | 29.54(4.46-62.12) | |
| Presence of DM | | | | | | | |
| | Absent | 9.11(4.06-18.07) | **0.002** | 2.79 (1.03-8.90) | **0.001** | 26.23(4.31-62.12) | 0.531 |
| | Present | 10.44(4.24-27.17) | | 3.21(1.63-10.40) | | 24.40(9.45-59.06) | |
| Hepatic steatosis | | | | | | | |
| | Absent | 9.07(4.06-23.53) | 0.163 | 3.05(1.06-7.60) | 0.740 | 25.23(4.31-62.12) | 0.849 |
| | Present | 9.80(4.29-27.17) | | 2.99(1.03-10.40) | | 24.26(9.45-55.44) | |
| LDL value | | | | | | | |
| | Normal | 8.93(5.35-14.59) | 0.235 | 2.59(1.06-4.83) | **0.030** | 23.10(4.31-56.47) | 0.290 |
| | High | 9.43(4.06-15.16) | | 3.07(1.12-5.36) | | 26.30(9.45-62.12) | |
| Pancreatic steatosis | | | | | | | |
| | Absent | 9.00(5.58-21.54) | 0.484 | 3.19(1.06-8.92) | 0.134 | 22.37(4.46-43.99) | 0.131 |
| | Present | 9.44(4.06-27.17) | | 2.97(1.03-10.40) | | 25.85(4.31-62.12) | |
| History of pancreatitis | | | | | | | |
| | Absent | 8.42(4.06-27.17) | 0.419 | 3.21(1.44-10.40) | **0.001** | 21.76(8.24-59.06) | **0.024** |
| | Present | 9.45(5.35-21.54) | | 2.93(1.03-8.92) | | 26.50(4.31-62.12) | |

Mann-Whitney U test. (EAT: epicardial adipose tissue).

**Table 2. Univariate and multivariate logistic regression analysis for predictors of diabetes mellitus.**

| Variables | Univariate | | Multivariate | |
|---|---|---|---|---|
| | OR (95% CI) | p | OR (95% CI) | p |
| EAT thickness at LAD level | 1.43 (1.14–1.80) | 0.002 | – | – |
| EAT thickness at RCA level | 1.21 (1.08–1.34) | 0.001 | 1.18 (1.06–1.32) | 0.002 |
| Subcutaneous fat at umbilical level | 0.99 (0.96–1.02) | 0.584 | – | – |
| Age | 1.04 (1.02–1.06) | 0.001 | 1.03 (1.01–1.06) | 0.003 |
| Sex | 0.62 (0.33–1.17) | 0.141 | – | – |
| Pancreatic steatosis | 0.52 (0.20–1.36) | 0.183 | – | – |
| LDL elevation | 0.49 (0.23–1.05) | 0.066 | – | – |
| Hepatosteatosis | 0.80 (0.43–1.49) | 0.478 | – | – |
| History of pancreatitis | 1.34 (0.67–2.68) | 0.415 | – | – |
| Pancreatic head density | 0.99 (0.98–1.01) | 0.832 | – | – |
| Pancreatic body density | 1.00 (0.99–1.01) | 0.983 | – | – |
| Pancreatic tail density | 1.00 (0.99–1.01) | 0.953 | – | – |

EAT: Epicardial adipose tissue; OR: Odds ratio; CI: Confidence interval.

## EAT and pancreatic steatosis/pancreatic density

No significant association was observedbetween pancreatic steatosis and EAT thickness at either the RCA or LAD levels (p > 0.05), and subcutaneous adipose tissue measurements at the umbilical level were also comparable between groups (Table 1). Similarly, pancreatic density values (head, body, or tail) were not associated with EAT thickness at either the

RCA or LAD levels (p > 0.05) (Table 3). Subcutaneous fat also showed no significant correlation with pancreatic density (Table 3).

## EAT and age

Correlation analysis demonstrated that age showed a weak but statistically significant positive correlation with EAT thickness at both the RCA (r = 0.161, p = 0.022) and LAD levels (r = 0.321, p < 0.001), indicating an age-related increase in epicardial adiposity (Table 3). Age also showed a weak but statistically significant negative correlation with subcutaneous adipose tissue thickness (r = −0.188, p = 0.008) (Table 3).

## EAT and pancreatitis history

We then evaluated the relationship between EAT and pancreatitis history, an important exploratory aim of the study. Patients with a history of pancreatitis had significantly higher LAD-EAT thickness (p = 0.001) and greater umbilical subcutaneous fat (p = 0.024), although RCA-EAT thickness was not significantly different (Table 1).

In logistic regression, pancreatic steatosis (OR = 6.91), LAD-EAT thickness (OR = 1.62), and pancreatic density parameters were significant univariate predictors of pancreatitis history. In multivariate analysis, only pancreatic steatosis (OR = 5.78) and LAD-EAT thickness (OR = 1.52) remained significantly associated with diabetes (Table 4).

**Table 3. Correlation of EAT thickness with age and pancreatic density measurements.**

| Variables | EAT thickness at RCA level | | EAT thickness at LAD level | | Subcutaneous fat at umbilical level | |
|---|---|---|---|---|---|---|
| | r | p | r | p | r | p |
| Age | 0.161 | **0.022** | 0.321 | **<0.001** | −0.188 | **0.008** |
| Pancreatic head density | −0.035 | 0.627 | 0.005 | 0.944 | −0.128 | 0.071 |
| Pancreatic body density | −0.043 | 0.547 | 0.009 | 0.899 | −0.114 | 0.107 |
| Pancreatic tail density | −0.016 | 0.824 | 0.036 | 0.617 | −0.076 | 0.286 |

Spearman correlation test. (EAT: Epicardial adipose tissue).

**Table 4. Univariate and multivariate logistic regression analysis of factors associated with pancreatitis history.**

| Variables | Univariate | | Multivariate | |
|---|---|---|---|---|
| | OR (95% CI) | p value | OR (95% CI) | p value |
| EAT thickness at LAD level | 1.62 (1.27–2.07) | <0.001 | 1.52 (1.17–1.97) | 0.002 |
| EAT thickness at RCA level | 1.04 (0.94–1.14) | 0.456 | – | – |
| Subcutaneous fat at umbilical level | 0.97 (0.94–0.99) | 0.040 | – | – |
| Age | 1.01 (0.99–1.03) | 0.378 | – | – |
| Sex | 0.87 (0.46–1.63) | 0.657 | – | – |
| Pancreatic steatosis | 6.91 (3.00–15.88) | <0.001 | 5.78 (2.39–13.93) | <0.001 |
| LDL elevation | 0.49 (0.23–1.05) | 0.066 | – | – |
| Hepatosteatosis | 0.86 (0.45–1.62) | 0.631 | – | – |
| DM presence | 1.34 (0.67–2.68) | 0.415 | – | – |
| Pancreatic head density | 1.02 (1.01–1.04) | 0.006 | – | – |
| Pancreatic body density | 1.02 (1.01–1.04) | 0.009 | – | – |
| Pancreatic tail density | 1.02 (1.01–1.04) | 0.011 | – | – |

EAT: Epicardial adipose tissue; OR: Odds ratio; CI: Confidence interval.

## Correlation between subcutaneous fat and EAT

We assessed whether abdominal subcutaneous adipose tissue reflected epicardial fat accumulation. Umbilical subcutaneous fat demonstrated a weak yet significant positive correlation with RCA-EAT (r = 0.153, p = 0.031), but not with LAD-EAT thickness (p = 0.779) (Table 5).

## Discussion

Our study investigated the relationship between EAT thickness and diabetes, dyslipidemia, hepatosteatosis, pancreatic steatosis, and pancreatitis. EAT and subcutaneous adipose tissue were found to be significantly greater in women, and EAT thickness increased in the presence of diabetes, particularly at the RCA and LAD levels. A distinctive aspect of this study is the evaluation of EAT thickness on CT scans performed for suspected pancreatitis, extending previous research beyond cardiovascular imaging cohorts. Multivariate analysis revealed that age and RCA-EAT were significantly associated with diabetes, Furthermore, high LDL levels were associated with LAD-EAT, and LAD-EAT and pancreatic steatosis were significantly associated with a history of pancreatitis.

EAT represents a dynamic visceral fat compartment positioned adjacent to the myocardium and beneath the visceral pericardium, chiefly surrounding the coronary arteries and occupying the atrioventricular and interventricular regions [1]. EAT consists of smaller adipocytes than other visceral fat depots, yet it shows greater fatty acid uptake and release [7]. Under normal physiological conditions, it serves to protect against elevated free fatty acid levels, provides a local energy source for the myocardium when needed, and serves as a thermogenic barrier through its brown-fat-like activity [8].

Although EAT has beneficial physiological roles, it may also release harmful pro-inflammatory cytokines that negatively affect coronary arteries and the myocardium [9]. Recent evidence has shown that EAT functions as a highly active endocrine organ, producing numerous bioactive adipokines [10].

Among the bioactive mediators secreted by EAT are pro-inflammatory and pro-atherogenic cytokines, including tumor necrosis factor-α (TNF-α), monocyte chemoattractant protein-1 (MCP-1), interleukin-6 (IL-6), nerve growth factor, resistin, visfatin, omentin, leptin, plasminogen activator inhibitor-1 (PAI-1), and angiotensinogen. In addition, it produces anti-inflammatory and anti-atherogenic factors such as adiponectin and adrenomedullin [11,12]. Increased EAT thickness is a known risk factor for cardiovascular diseases, contributing to chronic vascular remodeling by promoting the migration of pro-inflammatory cells in conditions such as obesity, insulin resistance, diabetes mellitus, and vascular injury [13]. Recent studies have also linked EAT to hepatic steatosis and LDL cholesterol levels, providing new diagnostic and therapeutic insights [1,2–4]. Consistent with these findings, our study also demonstrated a significant association between elevated LDL levels and increased LAD-EAT thickness. However, unlike some previous reports, we did not observe a significant association between hepatic steatosis and EAT thickness; this may reflect differences in patient selection, imaging methodology, or disease severity.

In this study, EAT thickness was found to be significantly higher in female participants across all measurement levels. While Yerramasu et al. reported greater EAT thickness in men, they also emphasized that this may vary depending on age, menopausal status, and hormonal differences [14]. Our findings may be explained by the older age and probable postmenopausal status of our female participants.

**Table 5. Correlation analysis between umbilical subcutaneous fat thickness and EAT measurements.**

| Variables | EAT thickness at RCA level | | EAT thickness at LAD level | |
|---|---|---|---|---|
| | r | p | r | p |
| Subcutaneous fat at umbilical level | 0.153 | **0.031** | −0.020 | 0.779 |

Spearman correlation test. (EAT: Epicardial adipose tissue).

Another important finding was the significant association between DM and EAT thickness at both RCA and LAD levels. This supports previous literature highlighting the relationship between EAT, insulin resistance, and glucose metabolism [1–3]. For example, Tarabay et al. reported significantly greater EAT thickness in type 2 diabetes patients in both parasternal echocardiographic views, which aligns with our findings [15]. In our study, both RCA and LAD EAT thickness were significantly associated with diabetes in univariate analysis. However, after adjusting for confounding factors, only age and RCA-EAT thickness remained significantly associated with diabetes, suggesting that increased RCA-EAT may reflect metabolic alterations linked to diabetes rather than indicating a causal relationship.

EAT thickness was also significantly associated with elevated LDL cholesterol levels, especially at the LAD level, corroborating the findings of Dönmez et al. in patients with familial hypercholesterolemia [4]. The relatively weaker correlation observed at the RCA level may reflect local hemodynamic and structural differences in fat distribution around the coronary arteries.

In addition, age was found to be positively related to EAT thickness, supporting previous reports such as Abbara et al., who noted increased EAT in individuals aged ≥65 years [16]. Although the correlations between age, EAT thickness, and subcutaneous adipose tissue were statistically significant, the effect sizes were weak (r = 0.15–0.32). Such small coefficients may have limited clinical relevance and should be interpreted with caution. These associations may reflect unmeasured confounding factors—particularly BMI or overall adiposity—rather than a direct physiological relationship. Since BMI was not available in our dataset, the potential influence of general adiposity cannot be excluded and represents an important limitation of the study.

No significant association was found between EAT thickness and hepatic or pancreatic steatosis in our study. Our results differ from those of Liu et al., who demonstrated a significant association between EAT and non-alcoholic fatty liver disease (NAFLD), as well as Kul et al., who demonstrated significantly increased EAT thickness in patients with non-alcoholic fatty pancreas [3–5]. The discrepancies may be due to our relatively smaller sample size.

In participants with a history of pancreatitis, subcutaneous fat thickness at the umbilical level was significantly increased, consistent with findings by Bükülmez et al., who associated both visceral and subcutaneous fat with acute pancreatitis and recurrence in pediatric patients [17]. In this study, pancreatitis history was significantly linked to EAT thickness measured at the LAD level, suggesting that pro-inflammatory cytokines released by EAT may contribute to systemic inflammation and increase the frequency of pancreatitis attacks. However, further research with larger patient populations is needed to confirm this finding.

Interestingly, although subcutaneous fat thickness in the umbilical region and pancreatic parenchymal densities (head, body, and tail) were significant in univariate analyses, these associations did not persist after adjustment in multivariate models. This suggests that both subcutaneous fat and pancreatic tissue characteristics may contribute to pancreatitis risk, but their effects are likely overshadowed by the stronger influence of visceral depots such as EAT and pancreatic steatosis. Furthermore, the lack of a significant association between RCA-EAT and a history of pancreatitis suggests a potentially site-specific role for LAD-EAT in reflecting inflammatory and metabolic processes.

This study has several limitations. First, the retrospective, single-center design may introduce selection bias and limits the generalizability of the results to broader populations. In addition, the study population consisted of patients referred with suspected pancreatitis, which may have influenced the clinical and metabolic profile of the cohort and should be considered when interpreting the generalizability of the findings. Another limitation is that, despite the moderate sample size, subgroup analyses such as history of pancreatitis or LDL status may still be insufficient. Another limitation is that EAT was measured at only two CT levels (mid-RCA and mid-LAD), which may not fully reflect total EAT volume or regional variability. Another is that hepatic and pancreatic steatosis were assessed using CT and MRI rather than histopathology, which may lead to diagnostic variability. Finally, the lack of inflammatory biomarkers limited our ability to directly explore the mechanistic pathways between EAT, systemic inflammation, and pancreatitis.

Another important limitation is the absence of key metabolic and clinical confounders such as body mass index (BMI), total body adiposity, hypertension status, and medication use (e.g., statins, antidiabetic agents). Because EAT and subcutaneous adipose tissue are strongly influenced by these factors, the associations observed with diabetes, LDL cholesterol, or pancreatitis may partly reflect unmeasured confounding rather than direct physiological relationships. The lack of these variables limits the ability to fully adjust our multivariate models, and therefore the reported associations should be interpreted cautiously. Future studies incorporating comprehensive metabolic profiling are needed to validate these findings.

## Conclusion

Epicardial adipose tissue functions as an active visceral fat store, influencing cardiovascular and metabolic health in both protective and pathological ways. Our study found that EAT thickness is significantly higher in female participants, likely influenced by age and menopausal status. We observed an association of increased EAT thickness with diabetes mellitus and elevated LDL cholesterol levels, suggesting a possible link with insulin resistance and atherogenesis. Age was positively correlated with EAT thickness. No significant relationship was found between EAT and hepatic or pancreatic steatosis, possibly due to sample size. However, increased EAT thickness in participants with a history of pancreatitis suggests a potential inflammatory link. These results highlight the need for prospective, multicenter studies incorporating comprehensive metabolic and inflammatory profiling to clarify the role of EAT in pancreatitis and cardiometabolic risk, and to determine whether EAT may have future value in risk stratification rather than immediate clinical application.

## Acknowledgments

The authors have no acknowledgments to declare.

## Author contributions

**Conceptualization:** Ece Zengin, Şehnaz Evrimler, Aybüke Uçgun, İhsaniye Süer Doğan, Mehmet Emir Çevik.

**Data curation:** Aybüke Uçgun, İhsaniye Süer Doğan.

**Formal analysis:** Ece Zengin, Şehnaz Evrimler, Aybüke Uçgun.

**Investigation:** Ece Zengin, İhsaniye Süer Doğan, Mehmet Emir Çevik.

**Methodology:** Ece Zengin, Aybüke Uçgun, İhsaniye Süer Doğan, Mehmet Emir Çevik.

**Project administration:** Ece Zengin, Şehnaz Evrimler.

**Resources:** Şehnaz Evrimler, Aybüke Uçgun, İhsaniye Süer Doğan.

**Software:** Mehmet Emir Çevik.

**Validation:** Ece Zengin, Mehmet Emir Çevik.

**Visualization:** Şehnaz Evrimler.

**Writing – original draft:** Aybüke Uçgun, İhsaniye Süer Doğan.

**Writing – review & editing:** Ece Zengin, Şehnaz Evrimler, İhsaniye Süer Doğan.

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
