## [Decision Letter · Decision Letter 0]

17 Nov 2025

Dear Dr. EVRİMLER,

Thank you for submitting your manuscript to PLOS ONE. After careful consideration, we feel that it has merit but does not fully meet PLOS ONE’s publication criteria as it currently stands. Therefore, we invite you to submit a revised version of the manuscript that addresses the points raised during the review process.

We look forward to receiving your revised manuscript.

Kind regards,

Eyüp Serhat Çalık

Academic Editor

PLOS ONE

Journal Requirements:

Additional Editor Comments:

I congratulate the esteemed authors for their efforts. The relationship between epicardial adipose tissue and various diseases has been investigated. Your study is generally well planned, and your manuscript is well structured and written. The manuscript was reviewed by two external reviewers, and their comments are below. Please provide point-by-point responses to the comments and make the necessary revisions to your manuscript. We look forward to receiving your revised manuscript. Best of luck.

Reviewers' comments:

Reviewer's Responses to Questions

**Comments to the Author**

1. Is the manuscript technically sound, and do the data support the conclusions?

Reviewer #1: Partly

Reviewer #2: Yes

2. Has the statistical analysis been performed appropriately and rigorously?

Reviewer #1: I Don't Know

Reviewer #2: No

3. Have the authors made all data underlying the findings in their manuscript fully available?

Reviewer #1: Yes

Reviewer #2: Yes

4. Is the manuscript presented in an intelligible fashion and written in standard English?

Reviewer #1: Yes

Reviewer #2: Yes

Reviewer #1: General Comments:

I found your study interesting and relevant, as it addresses the role of epicardial adipose tissue in cardiometabolic health. You clearly put effort into collecting and analyzing the data, and the manuscript is generally well structured. While reading, I paid close attention to how your results link to your study objectives and to the potential limitations of your cohort and methods. I have several comments that I believe could help clarify the interpretation of your findings, improve transparency, and make the manuscript stronger and more convincing for readers

Specific Comments:

Point1: I found the introduction rather weak in terms of focus and logical flow.You mention several conditions (hepatosteatosis, diabetes, pancreatitis, subcutaneous fat), but, there is no clear narrative connecting these elements. It feels as if several conditions were listed without a central rationale explaining why they were all included in the same study... The link between EAT and pancreatitis, in particular, seems weak and needs stronger biological justification. Since EAT is a cardiac fat depot, the biological pathway by which it could influence pancreatic inflammation is not evident. You might consider citing mechanistic or imaging studies, if available, or explaining your hypothesis more clearly. Also, stating that ""the link is not conclusive"" doesn’t clearly define the research gap. Readers need to understand waht exactely is uncertain or conflicting in the literature. I also noticed you didn’t refer to any existing studies with similar objectives or findings. Without that, it’s hard to see how your work differs from what’s already been done or what new aspect it brings. I suggest you restructure this section to highlight one clear gap, explain why these associations are relevant, and end with a precise hypothesis connecting EAT to the selected metabolic factors.

Point2: I think the methods section needs much more detail. While you report sample size and sex in the results section, it’s important to include this information in methods along with how participants were selected and the inclusion//exclusion criteria. Without that, it’s hard to know if your results are reliable or representative. Please consider adding these details so we can understand the conditions of your study and how representative your population is. Also, it would also help to clarify whether potential confounders like age, sex, or BMI were considered, as this is important for interpreting your results.

Point3 : The results section is detailed, but it feels somewhat descriptive and difficult to follow in relation to your study objectives. You report many associations, yet it’s not always clear which ones are most central to your objective. I suggest reorganizing the results according to your main aims (for example, EAT and diabetes, EAT and pancreatitis, etc.) and adding short linking sentences to guide the reader through the key findings. This would make the section clearer and more coherent.

Point4: I see that several of your results are statistically significant, but when I look at the actual correlation values. For example, between age, EAT, and subcutaneous fat (r = 0.15–0.32). These relationships appear quite weak. Even with low p values, such small effect sizes may not have much clinical meaning. I think it would really strengthen your paper if you could comment on whether these findings are meaningful in practice, or if they might simply reflect confounding factors like BMI or general adiposity.

point5: I noticed that some important potential confounders, such as BMI, overall body fat, hypertension, or medication use, don’t seem to have been accounted for in your analyses. Since EAT and subcutaneous fat are strongly influenced by these factors, the associations you report with diabetes, pancreatitis, or LDL cholesterol could be affected. Could you clarify whether these variables were available, and if so, consider including them in your multivariate models??? If not, a discussion of the possible impact of unmeasured confounding would help strengthen the study.

point6: In the regression analyses, it’s not clear how you decided which variables to include in the multivariate models. Were variables selected based on univariate significance (e.g.,, p < 0.1), or were all measured variables considered???

Point7: I see that you mention the retrospective design as a limitation, which is good. However, some of the language in the results and discussion, such as "independent predictors"" implies causality. Since causality cannot be established in a retrospective study, I suggest clarifying this in the text and using more neutral language (e.g.. "variables significantly associated with…") to avoid overinterpretation.

Point8: The prevalence of pancreatitis (73.5%%) and diabetes (31.4%) in your cohort seems unusually high compared with the general population. Could this reflect a selection bias; for example, if the sample mainly includes patients undergoing CT for pancreatic evaluation?? Could you clarify your sampling source and whether this distribution is representative of the broader population you aim to study..

Reviewer #2: The manuscript has potential but would require major revision before it could be suitable for publication. Please address the following comments in the manuscript:

The abstract lacks a clear problem statement as it lists results but does not sufficiently justify why this research matters or what specific knowledge gap it addresses. The final sentence should highlight the broader relevance rather than repeat outcomes.

The research gap is weakly articulated in the introduction. While prior work is mentioned, the authors do not convincingly explain what was missing in existing literature that this study uniquely addresses.

The rationale for selecting the study design is limited. The manuscript does not explain why the chosen population, tools, and parameters are most appropriate to answer the stated research question.

Methodology lacks detail and reproducibility in some areas. Key procedural specifics (e.g., selection criteria, sample handling, or validation of measurement tools) are missing or insufficiently described. This may prevent replication.

The study seems observational/descriptive, yet some statements imply causal relationships. These need to be toned down or reframed as associations only.

Sample size justification is not clearly provided. A power analysis or rationale for participant numbers would strengthen the manuscript.

The statistical analysis section is too brief. The authors should clarify corrections for multiple testing, how missing data were handled, and whether data met normality assumptions before applying parametric tests.

Results presentation feels fragmented. While data are provided, there is limited integration of findings into a coherent narrative. Tables/figures could be better linked in the text with clearer interpretation rather than simple reporting.

Tables and figures appear overly descriptive. Several contain too much raw numerical detail without clear summaries or take-home messages. They also need clearer captions that explain what readers should focus on.

Discussion largely restates results rather than interpreting them. There is little critical reflection on why certain outcomes emerged, how they compare to similar studies, or what mechanisms may be involved.

Limitations are acknowledged but underdeveloped. Key issues such as potential confounders, selection bias, and measurement limitations deserve a more honest discussion.

The manuscript overextends its conclusions. The findings support limited association but not practical recommendations. The authors should temper claims to match the strength of their data.

The implications for policy, clinical relevance, or future research are not clearly drawn. The conclusion could be more forward-looking, stating how these results should guide subsequent work.

Language is generally understandable, but certain paragraphs feel stiff and overly technical, reducing flow. Some grammatical issues remain; the manuscript would benefit from careful editing for clarity and conciseness.

Overall novelty appears modest. The study adds incremental information rather than a fundamentally new insight. The authors should emphasize what is genuinely new i.e., population studied, new analytical approach, or unique findings if applicable.

**Do you want your identity to be public for this peer review?** For information about this choice, including consent withdrawal, please see our Privacy Policy

Reviewer #1: No

Reviewer #2: No

---

## [Author Response · Author response to Decision Letter 1]

8 Jan 2026

We would like to thank the Editor and the reviewers for their careful evaluation of our manuscript and for their constructive and insightful comments. We have revised the manuscript thoroughly in response to all suggestions, which have helped improve the clarity, methodological transparency, and overall quality of the study. Below, we provide a point-by-point response to each comment, and all changes made to the manuscript are indicated with corresponding line numbers.

RESPONSE TO REVIEWERS

Reviewer 1.

Point 1. I found the introduction rather weak in terms of focus and logical flow. You mention several conditions (hepatosteatosis, diabetes, pancreatitis, subcutaneous fat), but, there is no clear narrative connecting these elements. It feels as if several conditions were listed without a central rationale explaining why they were all included in the same study... The link between EAT and pancreatitis, in particular, seems weak and needs stronger biological justification. Since EAT is a cardiac fat depot, the biological pathway by which it could influence pancreatic inflammation is not evident. You might consider citing mechanistic or imaging studies, if available, or explaining your hypothesis more clearly. Also, stating that ""the link is not conclusive"" doesn’t clearly define the research gap. Readers need to understand waht exactely is uncertain or conflicting in the literature. I also noticed you didn’t refer to any existing studies with similar objectives or findings. Without that, it’s hard to see how your work differs from what’s already been done or what new aspect it brings. I suggest you restructure this section to highlight one clear gap, explain why these associations are relevant, and end with a precise hypothesis connecting EAT to the selected metabolic factors.

Thank you very much for this insightful and constructive feedback. In line with your suggestions, we substantially revised the final paragraph of the Introduction to improve coherence, strengthen the biological rationale, and clearly articulate the research gap. First, we clarified that although EAT is anatomically a cardiac fat depot, it acts as an active visceral adipose tissue with systemic inflammatory effects and a strong correlation with total visceral fat, making its potential association with hepatic and pancreatic fat biologically plausible. We also added mechanistic justification by referencing the proinflammatory cytokines secreted by EAT and explaining how ectopic fat depots—including hepatic and pancreatic fat—share common metabolic and inflammatory pathways. Second, we specified that the relationships between EAT and pancreatic fat or pancreatitis history have not been systematically examined, and that existing literature on EAT and hepatosteatosis provides inconsistent results, thereby defining the precise gap our study addresses. Finally, we added a clearer rationale for why these metabolic factors were included together and concluded the Introduction with a precise, hypothesis-driven statement. All revisions have been incorporated into the manuscript and marked accordingly (Lines 61-73).

Point 2: I think the methods section needs much more detail. While you report sample size and sex in the results section, it’s important to include this information in methods along with how participants were selected and the inclusion//exclusion criteria. Without that, it’s hard to know if your results are reliable or representative. Please consider adding these details so we can understand the conditions of your study and how representative your population is. Also, it would also help to clarify whether potential confounders like age, sex, or BMI were considered, as this is important for interpreting your results.

Thank you very much for this valuable comment. We have substantially revised the Methods section to address your concerns and improve clarity and reproducibility. Specifically, we added detailed descriptions of the study population, including how patients were selected, the time frame of imaging, and the inclusion of abdominal CT/MRI examinations performed for suspected pancreatitis. We also incorporated explicit inclusion and exclusion criteria and clarified that age, sex, and BMI were collected as potential confounders relevant to ectopic fat distribution. In addition, we expanded the descriptions of the imaging measurements for EAT thickness, hepatosteatosis, and pancreatic fat, and clarified that all assessments were performed independently by two radiologists. These revisions ensure a complete and transparent methodological framework. All changes have been incorporated into the manuscript and marked accordingly (Lines 83-90 and 104-106).

Point 3 : The results section is detailed, but it feels somewhat descriptive and difficult to follow in relation to your study objectives. You report many associations, yet it’s not always clear which ones are most central to your objective. I suggest reorganizing the results according to your main aims (for example, EAT and diabetes, EAT and pancreatitis, etc.) and adding short linking sentences to guide the reader through the key findings. This would make the section clearer and more coherent.

Thank you for this valuable comment. We have substantially reorganized the Results section to improve clarity and alignment with our study aims. The findings are now presented under clear thematic subsections (e.g., EAT and Diabetes Mellitus, EAT and Hepatic Steatosis, EAT and Pancreatitis History), and brief linking sentences were added to guide the reader through the key results. These revisions improve the overall readability and coherence of the Results section. All changes have been incorporated into the revised manuscript (Lines 134-204).

Point 4: I see that several of your results are statistically significant, but when I look at the actual correlation values. For example, between age, EAT, and subcutaneous fat (r = 0.15–0.32). These relationships appear quite weak. Even with low p values, such small effect sizes may not have much clinical meaning. I think it would really strengthen your paper if you could comment on whether these findings are meaningful in practice, or if they might simply reflect confounding factors like BMI or general adiposity.

Thank you for this insightful observation. We agree that although several correlations reached statistical significance, their effect sizes were small (r = 0.15–0.32). To address this, we expanded the Discussion to clarify that these weak associations may have limited clinical relevance. We also added commentary noting that unmeasured confounders—particularly BMI and overall adiposity—may partially explain these relationships. The revised text now emphasizes the need for cautious interpretation and highlights this as an important limitation. These additions have been incorporated into the updated manuscript (Lines 257-263).

Point 5: I noticed that some important potential confounders, such as BMI, overall body fat, hypertension, or medication use, don’t seem to have been accounted for in your analyses. Since EAT and subcutaneous fat are strongly influenced by these factors, the associations you report with diabetes, pancreatitis, or LDL cholesterol could be affected. Could you clarify whether these variables were available, and if so, consider including them in your multivariate models??? If not, a discussion of the possible impact of unmeasured confounding would help strengthen the study.

Thank you for this insightful comment. Unfortunately, BMI, total body fat, hypertension status, and detailed medication history were not available in the retrospective dataset. We fully agree that these variables are important determinants of EAT and subcutaneous adiposity, and their absence may have introduced unmeasured confounding. To address this concern, we added a paragraph in the Discussion acknowledging this limitation and clarifying that the associations observed—particularly with diabetes, LDL cholesterol, and pancreatitis—should be interpreted with caution. We also highlighted the need for future prospective studies including comprehensive metabolic data (Lines 283- 287 and 293-301).

Point 6: In the regression analyses, it’s not clear how you decided which variables to include in the multivariate models. Were variables selected based on univariate significance (e.g.,, p < 0.1), or were all measured variables considered???

Thank you for this important comment. In our analysis, all measured variables were first entered into the univariate logistic regression. Only variables that showed statistical significance in the univariate analysis (p < 0.05) were subsequently included in the multivariate model. This stepwise approach was chosen to avoid overfitting and to ensure that only meaningful predictors were retained. Age and sex were kept in the model as clinically essential covariates. We have clarified this procedure in the Methods section (Lines 127-131).

Point 7: I see that you mention the retrospective design as a limitation, which is good. However, some of the language in the results and discussion, such as "independent predictors"" implies causality. Since causality cannot be established in a retrospective study, I suggest clarifying this in the text and using more neutral language (e.g.. "variables significantly associated with…") to avoid overinterpretation.

Thank you for this important comment. We fully agree that causal interpretations cannot be made in a retrospective study. In accordance with your suggestion, we carefully reviewed the Results and Discussion sections and revised the wording to avoid causal language. Terms such as “independent predictor” were replaced with neutral expressions including “significantly associated with” reflecting statistical relationships rather than causality. These revisions ensure that the findings are interpreted appropriately within the limitations of the study design (Lines 32-39, 162, 213-217).

Point8: The prevalence of pancreatitis (73.5%%) and diabetes (31.4%) in your cohort seems unusually high compared with the general population. Could this reflect a selection bias; for example, if the sample mainly includes patients undergoing CT for pancreatic evaluation?? Could you clarify your sampling source and whether this distribution is representative of the broader population you aim to study..

Thank you for this important observation. The high prevalence of pancreatitis and diabetes in our cohort indeed reflects a targeted sampling strategy rather than the general population distribution. As one of the predefined aims of the study was to examine the relationship between EAT and pancreatitis, patients who underwent abdominal CT/MRI due to clinical suspicion of pancreatitis were intentionally included. Therefore, the sample is enriched with individuals evaluated for possible pancreatitis, which explains the higher prevalence. We have clarified this point in the Methods section to avoid misinterpretation (Lines 83-86).

Reviewer 2.

Point 1. The abstract lacks a clear problem statement as it lists results but does not sufficiently justify why this research matters or what specific knowledge gap it addresses. The final sentence should highlight the broader relevance rather than repeat outcomes.

Thank you for your comment. We revised the abstract to clearly state the knowledge gap regarding the unclear relationship between EAT, pancreatic steatosis, and pancreatitis, and to justify the clinical relevance of the study. We also adjusted the final sentence to emphasize the broader significance rather than repeating results. We believe these changes enhance clarity and focus (Lines 18-21, 32-33, 35-39).

Point 2. The research gap is weakly articulated in the introduction. While prior work is mentioned, the authors do not convincingly explain what was missing in existing literature that this study uniquely addresses.

Thank you for this insightful comment. We revised the Introduction to more clearly define the research gap. The updated text now explicitly highlights: the lack of studies examining the relationship between EAT and pancreatic fat or pancreatitis, inconsistent evidence regarding the association between EAT and hepatosteatosis, and the uncertainty about whether EAT reflects only cardiac adiposity or broader systemic ectopic fat accumulation. These revisions strengthen the rationale and clarify the unmet need addressed by our study (Lines 61-73).

Point 3. The rationale for selecting the study design is limited. The manuscript does not explain why the chosen population, tools, and parameters are most appropriate to answer the stated research question.

Thank you for the comment. We clarified in the Methods why this study population and imaging parameters were appropriate. Abdominal CT/MRI enabled simultaneous evaluation of EAT, hepatic steatosis, pancreatic fat, and pancreatitis, directly aligning with our research aims. We also specified that patients imaged for suspected pancreatitis were intentionally included because assessing the EAT–pancreatitis relationship was one of the study objectives (Lines 83-86).

Point 4. Methodology lacks detail and reproducibility in some areas. Key procedural specifics (e.g., selection criteria, sample handling, or validation of measurement tools) are missing or insufficiently described. This may prevent replication.

Thank you for your constructive comment. We expanded the Methods section to provide full reproducibility, including detailed inclusion/exclusion criteria, rationale for the study population, CT/MRI measurement protocols and HU-based definitions (Lines 86-90).

Point 5. The study seems observational/descriptive, yet some statements imply causal relationships. These need to be toned down or reframed as associations only.

Thank you for this important remark. We carefully revised the manuscript to remove any causal wording. All expressions suggesting prediction or causality were replaced with neutral, association-based terminology (e.g., “significantly associated with”). The revised text now accurately reflects the observational nature of the study (Lines 163, 194-195, 216-217, 248-252).

Point 6. Sample size justification is not clearly provided. A power analysis or rationale for participant numbers would strengthen the manuscript.

Thank you for your constructive comment. Because this was an exploratory, retrospective study, no formal a priori sample size calculation was performed. The sample size was determined by the number of consecutive patients who met the inclusion criteria during the study period, which we considered sufficient to explore the hypothesized associations; however, future prospective studies with formal power calculations are warranted.

Point 7. The statistical analysis section is too brief. The authors should clarify corrections for multiple testing, how missing data were handled, and whether data met normality assumptions before applying parametric tests.

We thank the reviewer for this comment. There were no missing data in the dataset. Normality assumptions were assessed prior to applying parametric tests, and the relevant details have now been clarified and explicitly stated in the statistical analysis section (Lines 118-132).

Point 8. Results presentation feels fragmented. While data are provided, there is limited integration of findings into a coherent narrative. Tables/figures could be better linked in the text with clearer interpretation rather than simple reporting.

Thank you for this valuable comment. We have revised the Results section to create a more coherent and integrated narrative. Data are now presented with clearer transitions, and tables are referenced at points directly relevant to their findings. Each table is accompanied by brief interpretative statements rather than simple numerical reporting (Lines 133-204).

Point 9. Tables and figures appear overly descriptive. Several contain too much raw numerical detail without clear summaries or take-home messages. They also need clearer captions that explain what readers should focus on.

Thank you for this helpful comment. In line with your suggestion, we revised the caption of Table 1 to clearly highlight its key findings. Additionally, the captions of all other tables were rewritten to be more explanat

---

## [Decision Letter · Decision Letter 1]

12 Feb 2026

THE ASSOCIATION BETWEEN EPICARDIAL ADIPOSE TISSUE THICKNESS AND DIABETES MELLITUS, HYPERLIPIDEMIA, HEPATOSTEATOSIS, PANCREATIC STEATOSIS AND PANCREATITIS

PONE-D-25-52339R1

Dear Dr. Evrimler,

We’re pleased to inform you that your manuscript has been judged scientifically suitable for publication and will be formally accepted for publication once it meets all outstanding technical requirements.

Kind regards,

Eyüp Serhat Çalık

Academic Editor

PLOS One

Additional Editor Comments (optional):

Reviewers' comments:

Reviewer's Responses to Questions

**Comments to the Author**

Reviewer #1: All comments have been addressed

Reviewer #2: (No Response)

2. Is the manuscript technically sound, and do the data support the conclusions?

Reviewer #1: Yes

Reviewer #2: (No Response)

3. Has the statistical analysis been performed appropriately and rigorously?

Reviewer #1: I Don't Know

Reviewer #2: (No Response)

4. Have the authors made all data underlying the findings in their manuscript fully available?

Reviewer #1: Yes

Reviewer #2: (No Response)

5. Is the manuscript presented in an intelligible fashion and written in standard English?

Reviewer #1: Yes

Reviewer #2: (No Response)

Reviewer #1: I appreciate the author's thorough and thoughtful responses to my previous comments. The revisions have significantly improved the focus of the introduction, the transparency of the Methods, and the organization of the results, and the limitaion are now appropriately discussed. I have no further comments. Only minor editorial ajustements, if any, may be handeled at at the discretion of the editor.

Reviewer #2: (No Response)

**Do you want your identity to be public for this peer review?** For information about this choice, including consent withdrawal, please see our Privacy Policy

Reviewer #1: No

Reviewer #2: **Yes:** Zargham Faisal

---

## [Editor Report · Acceptance letter]

PONE-D-25-52339R1

PLOS One

Dear Dr. EVRİMLER,

I'm pleased to inform you that your manuscript has been deemed suitable for publication in PLOS One. Congratulations! Your manuscript is now being handed over to our production team.

Kind regards,

on behalf of

Dr. Eyüp Serhat Çalık

Academic Editor

PLOS One